



# Relationships Between Surface Fluxes and Boundary Layer Dynamics: Statistics at the Land-Atmosphere Feedback Observatory (LAFO)

**Syed Abbas, Andreas Behrendt, Oliver Branch, Volker Wulfmeyer**

Institute for Physics and Meteorology, University of Hohenheim, Stuttgart, Germany

*Correspondence to:* Syed Abbas (syed_saqlain.abbas@uni-hohenheim.de)

**Abstract.** We used a combination of two Doppler lidars (DLs) and an eddy covariance station at the Land-Atmosphere Feedback Observatory (LAFO), Stuttgart, Germany, to investigate relationships between surface fluxes, convective boundary layer (CBL) height, and profiles of vertical wind variance, horizontal wind variance and turbulent kinetic energy (TKE). One

DL was operated in vertical-pointing mode and the other in six-beam scanning mode. Daytime statistics were derived from 20 convective days from May to July 2021. In this data set, the mean CBL height $\langle \overline{z_i} \rangle$ showed a maximum of $(1.53 \pm 0.07)$ km between 13:00 and 14:00 UTC, which is about 1.5 to 2.5 hours after local noon. We found counterclockwise hysteresis patterns between the CBL height and the surface fluxes. In the development phase, these relationships were approximately linear. In the early afternoon, the relationships reached a peak phase with both large fluxes and high values of $\langle \overline{z_i} \rangle$. At 12:00 UTC, just

after local noon, the maximum values of vertical, horizontal, and total TKE were $0.55$ m$^2$s$^{-2}$, $1.26$ m$^2$s$^{-2}$ and $1.71$ m$^2$s$^{-2}$ at heights of $(0.30 \pm 0.06)\langle \overline{z_i} \rangle$ , $(0.56 \pm 0.06)\langle \overline{z_i} \rangle$, and $(0.40 \pm 0.06)\langle \overline{z_i} \rangle$, respectively. In the decay phase in the late afternoon, the relationships show non-linear patterns with larger values of $\langle \overline{z_i} \rangle$ for the same surface fluxes than in the morning. Furthermore, we show relationships between the vertical and horizontal components and total TKE.

## 1 Introduction

The atmospheric boundary layer (ABL) is the lower part of our atmosphere which is directly influenced by the land surface exchanges of energy, momentum, and mass (Seibert, 2000; Stull, 1988; Yi, 2001). ABL evolution depends on processes occurring at spatial scales from tens of meters to a few km and temporal scales from seconds to hours. In daytime, the ABL becomes convective and turbulent processes dominate. Typically, this transition to a convective boundary layer (CBL) follows a daily cycle with a morning transition to a maximum. This maximum appears slightly after local noon, followed by a decay

phase in late afternoon. Turbulence in the CBL is driven primarily by surface heating (Manninen et al., 2017), wind shear, and tropospheric entrainment (Wulfmeyer et al., 2016). The growth rate of the CBL is a result of the net interaction of these phenomena, which are in turn influenced by humidity (e.g., Salvucci and Gentine, 2013), advected air mass properties (e.g., Kossmann et al., 1998), cloud cover (e.g., Raupach, 1998), vegetation characteristics, and soil moisture (e.g., Shi et al., 2013; Vogel et al., 2017). Thus, the surface and CBL are inextricably linked in a complex way by nonlinear feedbacks. This land-

atmosphere (LA) bi-directional interactions involve radiative, atmospheric, and soil/vegetation fluxes and states. These interactions lead to a diurnal cycle of surface fluxes (Wulfmeyer et al., 2018). The complex relationship between soil moisture





and precipitation (Seneviratne et al., 2010; Guillod et al., 2015; Santanello et al., 2018) influences the turbulent fluxes at the surface (Ek and Holtslag, 2004) and influence CBL dynamics (Taylor et al., 2012). Additionally, the transpiration of vegetation during summer in the mid latitudes also influences CBL dynamics. Understanding these interactions and feedbacks is vital for

better performance of numerical climate and weather models.

For studies of turbulent transport processes in the CBL, we analyse here vertical wind variance and turbulent kinetic energy (TKE) profiles. Vertical wind variance determines spatial and temporal fluctuations in wind speed, which helps to identify whether atmospheric conditions are stable or convective. TKE is related to turbulent transport of momentum, heat, and water vapor in the atmosphere (e.g., Stull 1988). In a turbulent flow, the size of eddies covers a wide range of length scales, fluctuating

at different frequencies (Tennekes and Lumley, 1972). The larger the eddy sizes, the larger the vertical wind variance and TKE (Wulfmeyer et al., 2016). Accurate parameterization of TKE is crucial for modelling turbulent processes in weather and climate models, directly affecting forecasts of atmospheric variables (Olson et al., 2019).

In recent years, Doppler lidar (DL) systems have become more reliable. The continuous operation of such systems enables us to collect long-term datasets of high spatiotemporal resolution for studying the statistics of CBL dynamics. In vertical staring

mode, with DLs, higher-order moments of vertical wind fluctuations like atmospheric vertical wind variance $\overline{w'^2}$ can be determined, including uncertainties (Lenschow et al., 2000, Wulfmeyer et al., 2016, Wulfmeyer et al., 2024). $\overline{w'^2}$ profiles exhibit a characteristic shape in the CBL with typically single maxima at around 0.4 $z_i$ (Lenschow et al., 1980), where $z_i$ is the CBL height. In a six-beam scanning mode, a DL enables us to compute simultaneously the profiles of mean horizontal wind (Newsom et al., 2017), TKE and momentum flux (Bonin et al., 2017) within the CBL.

Renner et al. (2019) presented the hysteresis effects between surface fluxes and incoming solar radiation are result of time dependent non-linear control of heat storage and energy redistribution in the CBL. The surface energy balance closure is fundamental to understanding LA interaction. This involves accurate assessment of turbulent fluxes, ground heat flux, and net radiation. Typically, the surface fluxes are calculated using the eddy covariance (EC) technique (e.g., Mauder et al., 2020). However, the EC technique encounters an energy balance closure problem where usually one or both of the surface turbulent

fluxes are underestimated. Current understanding is that the missing energy is because of quasi-stationary micro- and mesoscale circulations generated by surface heterogeneities (Mauder et al., 2020). These features lead to dispersive fluxes that are not captured by the traditional EC methods. Typically, the discrepancy difference between net radiation ($Q_n$) and ground heat flux ($G$) is on the order of 10 to 30 percent of the sum of surface sensible ($S$) and latent heat flux ($L$) (Foken et al., 2011). The relationship between CBL and surface fluxes is crucial for improving predictions within weather and climate models (Chu

et al., 2022), and transport of tracers and pollutants (Lawston-Parker et al., 2021). The land surface properties influence the partitioning of turbulent fluxes (Chen and Lo, 2023). The CBL growth rate is directly dependent on land surface conditions and this coupling is influenced by feedback between surface energy balance and entrainment of dry and warm air from the free troposphere.





The Land-Atmosphere Feedback Observatory (LAFO) of University of Hohenheim in Stuttgart, Germany, deploys a
synergistic sensor network to measure simultaneously high-resolution state variables from the bedrock to the lower free
troposphere (Späth et al., 2023). LAFO is a prototype of the GEWEX LAFOs (GLAFOs), a project of the Global Land
Atmosphere System Study (GLASS) panel (Wulfmeyer et al., 2020). LAFO consists of a suite of active remote sensing
systems, along with a soil temperature and moisture sensor network, two eddy covariance (EC) stations complemented by
intensive crop observations. In this paper, we employ the combined data from two DLs, one EC station, and selected canopy
measurements to study the CBL properties.

The objective of this paper is to study the relationships between $z_i$, $\overline{w'^2}$, TKE and surface fluxes within an evolving CBL. To
achieve this, we used three months of LAFO data from May to July 2021.

The structure of this paper is as follows: After this introduction, we discuss the instrumentation and meteorological conditions
during the analysis period in section 2 and the methodology in section 3. In section 4, we present our findings on key
relationships between CBL height, surface fluxes, vertical wind variance, TKE. Finally, we summarize our results and put the
scientific findings in context and outline potential avenues for further research.

## 2 Observation Site, Instruments, and Selected Data Period

### 2.1 The Land-Atmosphere Feedback Observatory (LAFO)

The Land-Atmosphere Feedback Observatory (LAFO, 48.714° N, 9.184° E) is located in south-western Germany, 7 km south
of central Stuttgart at the University Hohenheim. The land cover around LAFO is primarily agricultural land, with maize,
rapeseed, wheat, triticale, barley, and other crops (Fig. 1). The LAFO experimental site is relatively flat, with an elevation
range of 390 to 420 m. The mean monthly temperature varies between 0 °C in winter and 18 °C in summer. The main purpose
of LAFO is to acquire continuous vertical and horizontal observations from the soil, land cover, and the atmosphere through a
unique synergy of in-situ and active remote sensing instruments. As part of this synergy, Doppler lidars (Halo Photonics), a
cloud Doppler radar (Metek GmbH), a water vapor differential absorption lidar (DIAL) (Späth et al., 2016), and a water vapor
and temperature rotational Raman lidar (RRL) (Lange et al., 2019) are deployed to observe land surface fluxes and flux
profiles, and other horizontal and vertical processes in the ABL (Wulfmeyer et al., 2016, Wulfmeyer et al., 2018).

Two observational strategies are employed at LAFO. The first is the systematic observation and archiving of continuous time
series of measured LA system variables and states to provide seasonal and multi-year statistics. This long-term instrumentation
has been operating continuously at LAFO since 2018. Further, these continuous datasets are then augmented during selected
'intensive observational periods (IOPs) of interest. A full description of LAFO and its instrumentation is given by Späth et al.
(2023).



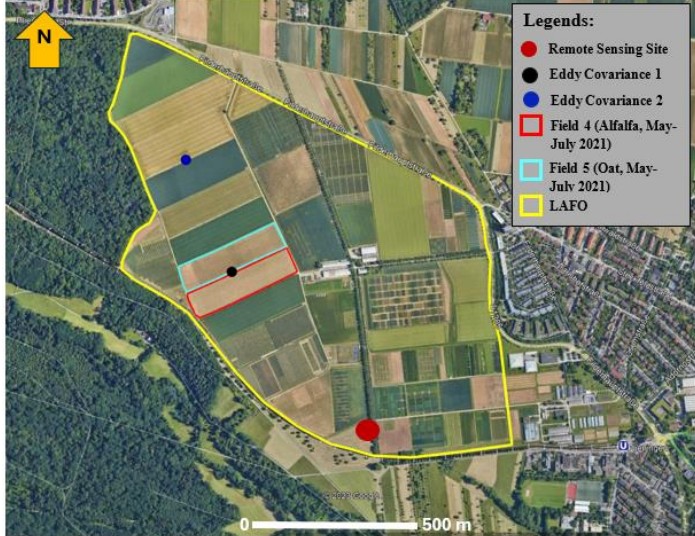

**Figure 1.** Satellite image (source: © Google Earth, 2022) of the Land Atmosphere Feedback Observatory (LAFO) showing
the positions of the remote sensing site where we deploy, among other instruments, two Doppler lidars, a Doppler cloud radar,
and thermodynamic lidars. The two eddy covariance stations (Eddy Covariance 1 and 2) are located at distances of about 650
m and 900 m, respectively, from the remote sensing site in a north-west direction. In the neighbouring fields of Eddy
Covariance 1 (Fields 4 and 5), alfalfa and oat plants were grown during the analysis period of this study in May to July 2021.

To investigate LA interaction processes, we employ among other instruments, a synergy of two DLs and one EC station at
LAFO. The vertical DL and the scanning DL provide us with data of radial wind velocity, signal-to-noise ratio (SNR) and
attenuated backscatter $\beta$ with high spatial and temporal resolutions. From these data, we get not only standard variables like
horizontal wind profiles, cloud base height (CBH) and the CBL height $z_i$ but can also derive turbulent variables like vertical
wind fluctuations and their high-order moments, turbulent kinetic energy (TKE), and momentum fluxes.
Together with turbulent flux data at the surface, we aim at improving our understanding of the vertical structure of turbulence
in the CBL, and its interaction with the land surface.

### 2.1.1 Eddy Covariance Stations

At LAFO, two EC stations continuously measure standard meteorological and high-resolution energy balance data. The high-
resolution flux measurements are collected using a Campbell Scientific CSAT3 sonic anemometer and a Licor 7500 gas
analyzer. From these 10 Hz data, 30-min surface energy fluxes are estimated using the EC method (e.g. Mauder et al., 2020).
Here we employ the TK3 software package developed by Mauder and Foken, (2015). For data quality control, we have used
the flag system of Mauder et al. (2013) for $\overline{Q_n}$, $\overline{S}$ and $\overline{L}$, and we exclude any data that falls outside of -100 W m$^{-2}$ to 800



W m$^{-2}$. For the full energy balance, upwelling and downwelling shortwave (SW) and longwave (LW) radiation components
are measured using a Hukseflux NR01 4-way radiometer. Ground heat flux is measured using three (averaged) Hukseflux
HFP01 plates at 8 cm depth, corrected for soil heat storage with change in soil temperature between 2 cm to 6 cm. In addition,
surface meteorological quantities are measured, such as air temperature, humidity, pressure, precipitation, and soil temperature
and volumetric water content. (Späth et al., 2023).

In this study, we used flux data from the EC station 1 at LAFO (see Fig. 1), closest to the position of the DLs and the DCR.
Our analysis shows that the footprint of Eddy Covariance 1 was always within neighbouring fields 4 and 5. Alfalfa was planted
around the second week of May 2021 in field 4, and oat was planted in the second week of March 2021 in field 5. In field 4,
the canopy height of alfalfa increased from 1 cm in mid-May to 2 cm by the end of May 2021. It showed significant growth in
June, increasing from 5 cm at the beginning of the month to 50 cm by the end. In the beginning of July 2021, its growth rate
continued at a gradual pace and reached a maximum height of 70 m by the end of July. In field 5, oats showed relatively rapid
growth from 18 cm in the beginning of May to 55 cm by the end of May 2021. In June, its canopy height increases from 65
cm to 100 cm. In July, the canopy height no longer changed until the end of the analysis period.

### 2.1.3 Vertical Pointing and Six-Beam Scanning Doppler Lidars

The two DLs were used for radial wind velocity measurement. Their working principle is based on the Doppler shift of near-
infrared laser light with a wavelength of 1.5 µm, backscattered from aerosol and cloud particles in the atmosphere (e.g., Berg
et al., 2017). These DLs were manufactured by Lumibird/Halo Photonics Ltd. (Pearson et al., 2009). One DL is operated
continuously in a vertical pointing mode with temporal/spatial resolutions of 1 s/30 m. The second DL is operated in a six-
beam stop-stare mode, meaning that the DL performs measurements with 5 beams, each with a constant elevation angle
typically 45° in 72° azimuth steps, as well as one vertically pointing beam. For each position, the measurement time is 10 s
with 30 m spatial resolution. This six-beam pattern is then repeated continuously. The time needed for one complete cycle is
about 90 s.

### 2.1.3 Doppler Cloud Radar

For identifying periods with precipitation, we employed a MIRA 35c vertically pointing Ka-band Doppler Cloud radar (DCR),
of Metek GmbH. The radar has a frequency 35.1 GHz frequency (Görsdorf et al., 2015). Due to its longer wavelength
compared to the DL, the radar signal is sensitive to cloud droplets, raindrops, and ice crystals. Consequently, it extends the
radial velocity measurements to the interior of clouds where the DL signal is strongly attenuated. Similarly to the
lidar, these are radial velocities.





### 2.1.4 Meteorological Conditions During the Analysis Period

Strong convective conditions occur on warm, mostly clear-sky days when thermally driven turbulence dominates. For this study, we selected only days with cloud cover <40% during daytime, to ensure that the CBL was indeed developed by convection.


The presence of precipitation was identified profile by profile using the DCR data, with the assumption that precipitation is occurring when negative radial velocities exceed ~-4 ms$^{-1}$. Whenever precipitation was detected between 06:00 and 19:00 UTC, that day was excluded from the analysis presented here. For the cloud identification and filtering, we applied the two following criteria: If the cloud cover was <40%, this day was classified as a convective day, and days with cloud cover >40%

were classified as cloudy days. The cloud occurrence is estimated for all days though a procedure explained by Newsom et al., (2019b). The CBH are identified by the heights of sharp gradients in the 1 s range-corrected SNR profiles of the vertical pointing DL. It is applied to those profiles of SNR for which the values of the backscatter coefficient $\beta$ exceeds $10^{-4}$ m$^{-1}$sr$^{-1}$. After detecting the CBH, the corresponding profiles from lidar vertical wind data are removed from the CBH point up to 3 km top. For our analysis, only those days are selected when cloud cover is <40%, otherwise the days were excluded.

After cloud and precipitation filtering, we have selected the following twenty days for the analysis: May 08, 09 and 31; June 01, 02, 14, 15, 16, 17, 18, 19, 21, 27, and 28; and July 02, 03, 10, 12, 18, and 31.

To analyse further the prevailing condition of these twenty days, we discuss the surface fluxes and temperature data measured at the closest EC station to the remote sensing site at LAFO. In Fig. 2, we show the flux data and temperature data averaged over all 20 selected days in the course of the days. In Fig. 2a and Fig. 2b, the mean net radiation $\langle \overline{Q_n} \rangle$ and the mean incoming

solar radiation $\langle \overline{Q_s} \rangle$ follow typical diurnal patterns. Before sunrise, $\langle \overline{Q_n} \rangle$ is negative while $\langle \overline{Q_s} \rangle$ is zero. During daytime, both are positive. Between 11:00 to 13:00 UTC, $\langle \overline{Q_n} \rangle$ and $\langle \overline{Q_s} \rangle$ are larger than 600 Wm$^2$ and 800 Wm$^2$, respectively. $\langle \overline{Q_n} \rangle$ and $\langle \overline{Q_s} \rangle$ reach their maximum values of (661±14) W m$^{-2}$ and (893±14) W m$^{-2}$, respectively, one hour after local noon at 12:30 UTC. In the afternoon, both $\langle \overline{Q_n} \rangle$ and $\langle \overline{Q_s} \rangle$ begin to decrease systematically and continue to decline until the evening. It is interesting to note that the variability of $\langle \overline{Q_n} \rangle$ and $\langle \overline{Q_s} \rangle$ is similar, lower during morning than in the afternoon. This is related to more

cumulus clouds at the CBL top in the afternoon than in the morning. The mean of the sensible heat flux $\langle \overline{S} \rangle$ shows a similar pattern to $\langle \overline{Q_n} \rangle$ and $\langle \overline{Q_s} \rangle$ (Fig. 2c). The maximum value of $\langle \overline{S} \rangle$ is (92±13) Wm$^{-2}$ at 12:00 UTC. In the afternoon, $\langle \overline{S} \rangle$ declines and reaches to negative values in the evening hours at 17:00 UTC. Interestingly, this happens already about 2h before $\langle \overline{Q_n} \rangle$ shows negative values. In Fig. 2d, the mean of the latent heat flux $\langle \overline{L} \rangle$ exhibits a similar pattern. Its maximum value is (403±20) W m$^{-2}$ at 12:30 UTC. Fig. 2e shows the maximum ground heat flux $\langle \overline{G} \rangle$ of (92.22±11) W m$^{-2}$ at 11:30 UTC around

local noon. $\langle \overline{G} \rangle$ remains positive between 06:00 to 18:00 UTC. The minimum value of $\langle \overline{T_{2m}} \rangle$ is 11.7 °C at 04:30 UTC before sunrise, while the maximum value is 25.2 °C at 16:00 UTC (Fig. 2f). $\langle \overline{T_{2m}} \rangle$ lags behind $\langle \overline{Q_n} \rangle$ as it is typical (Renner et al., 2019, Wu et al., 2023).



**Figure 2.** Averages of the 30-minute data of the 20 convective days from May to July 2021 measured at Eddy Covariance 1: (a) net radiation $\langle \overline{Q_n} \rangle$, (b) incoming solar radiation $\langle \overline{Q_s} \rangle$, (c) sensible heat flux $\langle \overline{S} \rangle$, (d) latent heat flux $\langle \overline{L} \rangle$, (e) ground heat flux $\langle \overline{G} \rangle$, and (f) temperature at 2 m height $\langle \overline{T_{2m}} \rangle$. Local noon at 11:30 UTC is marked with spheres. Error bars show 1-sigma standard deviations. The maximum and minimum values are also shown.



## 3 Methodology

### 3.1 Vertical Wind Variance

The vertical pointing DL measures the profiles of vertical wind $w(x,t)$ along the line of sight (LOS) at different altitudes within the CBL as

$$w(x,t) = \overline{w}(x) + w'(x,t) + \varepsilon(x,t) \ , \tag{1}$$

where $\overline{w}(x)$ is the mean vertical wind, $w'(x,t)$ is the vertical wind fluctuation, and $\varepsilon(x,t)$ the instrumental noise at time $t$ and height $x$. Lenschow et al. (2000) introduced an efficient technique to estimate higher-order-moments of lidar data by separating

$\overline{w'^2}$ from $\varepsilon(x,t)$. This technique, known as autocovariance technique, is based on the fact that $w'(x,t)$ is correlated in time whereas $\varepsilon(x,t)$ is not. Therefore, $\varepsilon(x,t)$ does not influence the autocovariance function ACF at time lags other than zero.

Before applying the ACF, we have performed gridding of $w(x,t)$ with 50 m spatial and 10 s temporal resolution. Then the time series of $w(x,t)$ were detrended with linear fits at each height. This detrending removes effects of large-scale advection and small scale trend within the diurnal cycle to focus on turbulence fluctuations (Muppa et al., 2016). In the subsequent phase,

$w(x,t)$ is averaged over 30 minutes obtaining $\overline{w}(x)$, which is then subtracted from $w(x,t)$:

$$w'(x,t) + \varepsilon(x,t) = w(x,t) - \overline{w}(x) \ , \tag{2}$$

In the next step, $w'(x,t)$ that exceed 4 standard deviations are discarded as outliers because they are due to large instrumental noise (Wulfmeyer et al., 2024). Then we derive the ACF function $M_{ij}$ of $w'(x,t)$.

The ACF of $w'(x,t) + \varepsilon(x,t)$ with time lag $\tau$ and time length $T$ for a given height $x$ (Lenschow et al., 2000) is

$$M_{ij}(\tau) = \frac{1}{T} \int_0^T [(w'(x,t) + \varepsilon(x,t)]^i [(w'(x,t+\tau) + \varepsilon(x,t+\tau)]^j \ dt \ , \tag{3}$$

The second order ACF, $M_{11}(\tau)$ at $\tau = 0$, is then

$$M_{11}(\tau = 0) = \frac{1}{T} \int_0^T [(w'(x,t) + \varepsilon(x,t)]^2 \ dt, \tag{4}$$

If the instrumental noise $\varepsilon(x,t)$ was 0, then $M_{11}(0) = \overline{w'^2}$. Since this is not the case for lidar measurements, we need to extract the noise contribution to the $M_{11}(\tau)$. This can be done because noise is not correlated on time and thus not present in $M_{11}(\tau)$ with $\tau = 0$.

$M_{11}(\rightarrow 0)$ is the extrapolation of the second order ACF to zero lag, which corresponds to $\overline{w'^2}$ (Lenschow et al., 2000). $\overline{\varepsilon^2}$ can be determined as

$$\overline{\varepsilon^2} = M_{11}(0) - M_{11}(\rightarrow 0) \ , \tag{5}$$

According to Monin and Yaglom (1971), $M_{11}(\tau)$ can be approximated with

$$M_{11}(\tau) = \overline{w'^2} - k_w \ \tau^{2/3} \ , \tag{6}$$

In Eq. (6), $k_w$ is the ACF coefficient for vertical wind.



In this study, we have used 15 lags of sampling rate 10 s covering 150 s to extrapolate the data to zero lag. We found that this number of fit lags is sufficient to resolve the inertial subrange. The inertial subrange is the time interval in which the turbulence scales are locally homogeneous and isotropic within the CBL (Wulfmeyer et al., 2016).

## 3.2 Turbulence Kinetic Energy

TKE is the overall energy per unit mass carried by turbulent eddies during convection. In boundary layer meteorology TKE is thus (Stull, 1988):

$$\text{TKE} = \frac{1}{2}\left(\overline{u'^2} + \overline{v'^2} + \overline{w'^2}\right),\tag{7}$$

Here, $\overline{u'^2}, \overline{v'^2}, \overline{w'^2}$ are the variances of the horizontal wind components $u$ and $v$ and the vertical component $w$.

We use the six-beam scanning technique to determine TKE, momentum flux, and horizontal wind profiles (Bonin et al., 2017). Reynolds stress is the fundamental turbulent quantity, which facilitates the turbulent transport of momentum within the CBL. Sathe et al. (2015b) showed that all the six components of the Reynolds stress tensor $R$ can be computed with this technique.

$$R = \begin{bmatrix} \overline{u'^2} & \overline{\langle u'v'\rangle} & \overline{\langle u'w'\rangle} \\ \overline{\langle v'u'\rangle} & \overline{v'^2} & \overline{\langle v'w'\rangle} \\ \overline{\langle w'u'\rangle} & \overline{\langle w'v'\rangle} & \overline{w'^2} \end{bmatrix},\tag{8}$$

In this matrix, the diagonal terms are the individual velocity variances whereas, the off diagonal terms represent the momentum fluxes. This method assumes that the atmosphere is horizontally homogeneous, and quasi-stationary, and the CBL is thus well-mixed over 30-minutes of each analysis period (Bonin et al., 2018).

We estimate the components of $R$ and subsequently the TKE profiles with the six-beam scanning DL as follows. We get a time series of instantaneous radial wind $v_{rn}(x,t)$ in six different directions. At first, each of the beams are linearly detrended over a fixed time window of 30 minutes.

Following this, the mean of each beam direction $\bar{v}_{rn}(x)$ over the same time window at each range gate height is computed. The fluctuations $v'_{rn}(x,t)$ containing instrumental noise $\varepsilon_{rn}$ can be written as:

$$v'_{rn}(x,t) + \varepsilon_{rn}(x,t) = \bar{v}_{rn}(x) - v_{rn}(x,t),\tag{9}$$

Next, we calculate the atmospheric variance $\overline{v'^2_{rn}}$ of each of the beam directions along with instrumental noise variance $\overline{\varepsilon'^2_{rn}}$. For this, we use the same way as described for the vertical wind in section 2.2.1.

Once the atmospheric variances are computed for each of the beams, the six unknown components of $R$ in a turbulent flow can be computed as:

$$R = M^{-1}A,\tag{10}$$

Where $M$ is a (6,6) matrix of coefficients based on different combinations of azimuth $\theta$ and elevation $\Phi$ (Bonin et al., 2018), and $A = \left[\overline{v'^2_{r1}}, \overline{v'^2_{r2}}, \overline{v'^2_{r3}}, \overline{v'^2_{r4}}, \overline{v'^2_{r5}}, \overline{v'^2_{r6}}\right]$ is a vector which represent the atmospheric variance at each beam direction. Hence, it is possible to derive the six unknown components of the Reynolds stress tensor separating the instrumental noise (Bonin et al.,





2018). Finally, total TKE $\overline{TKE_{tot}}$ is calculated by taking the sum of the atmospheric variances of fluctuations of the three wind components $u'$, $v'$, $w'$ obtained from $R$ as described in Eq. (7).

### 3.2 Convective Boundary Layer Depth

$\overline{w'^2}$ profiles derived from vertical pointing DL data provide an estimation of the CBL depth during daytime (Tucker et al., 2009). However, Bonin et al., (2018) demonstrated that this is not valid in all cases, especially during the presence of internal gravity waves at the top of the CBL. In such a case, it is necessary to perform high-frequency DL data filtering because otherwise $z_i$ may be overestimated. Bonin et al. (2018) developed a fuzzy logic algorithm for mixing height determination by integrating variances obtained from velocity-azimuth display (VAD), range height indicator (RHI), vertical stare and horizontal

shallow stare scans. This technique is adaptable to the user's need and DL data type. The maximum height of surface-based turbulence can be estimated using $\overline{w'^2}$ profiles obtained from the DL data.

    We included $\overline{w'^2}$ values across all heights for the CBL height estimation if $\overline{\varepsilon^2}$ <1 ms$^{-2}$, otherwise, $\overline{w'^2}$ was not considered. We applied a half trapezoidal membership function to transform $\overline{w'^2}$ values into fuzzy values between 0 (for non-turbulent conditions) to 1 (for strongly turbulent conditions) (Bonin et al., 2018). Here, we set two thresholds to detect turbulence: 0.04

m$^2$s$^{-2}$ as a lower and 0.2 m$^2$s$^{-2}$ as upper limit. These selected thresholds seem to fit best for our dataset. It is interesting to note that this algorithm not only functions well in clear sky conditions but also with cloud cover and precipitation. However as already continued above, we considered only days with less than 40% cloud cover without precipitation for our analysis.

    In Fig. 3, we show the instantaneous CBL depth $z_i$ , $w(x,t)$ and $\overline{w'^2}$ in the period of 14 to 19 June 2021 as an example. During these six days, the cloud cover was always <40%. Each day, the growing, the peak and the decaying phases of the CBL can be

seen clearly in the vertical wind data.





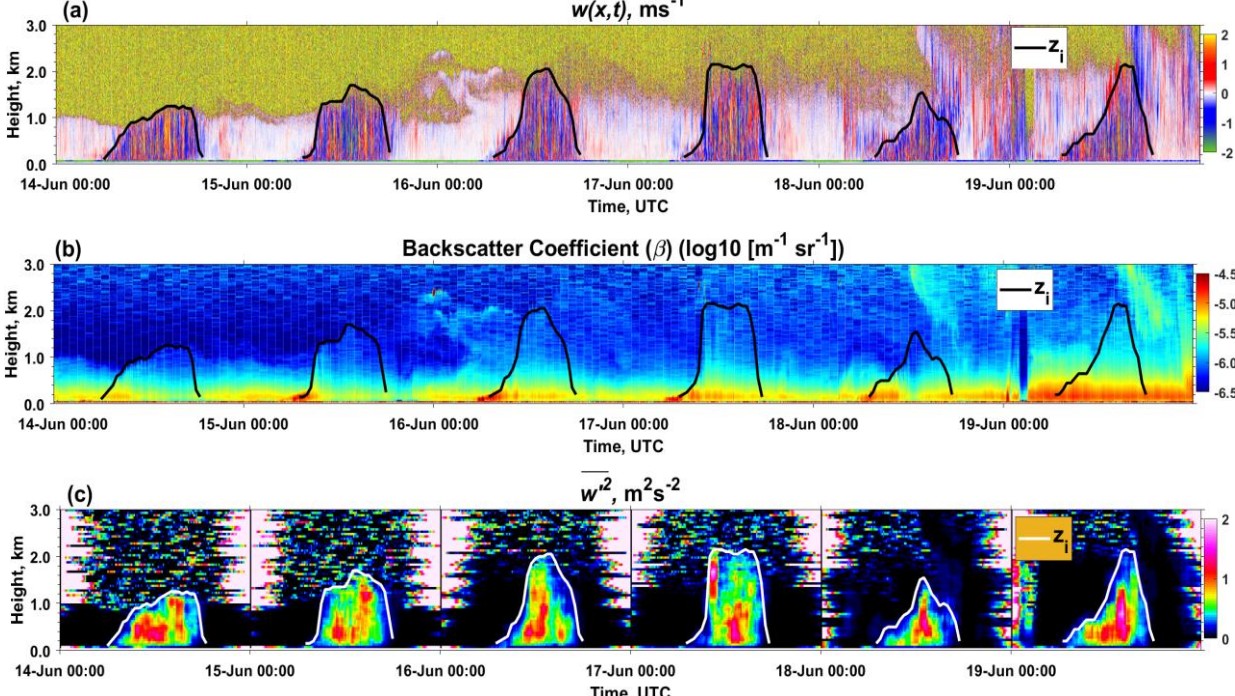

**Figure 3.** Time height cross sections of (a) vertical wind velocity $w(x, t)$ and (b) backscatter coefficient $\beta$ with resolutions of

10 s and 30 m from 14 to 19 June 2021 with instantaneous CBL heights $z_i$. (c) Atmospheric vertical wind variance $\overline{w'^2}$ with resolutions of 10 minutes and 50 m for the same period. The top of the convective boundary layer $z_i$ is shown in all plots for comparison.





The daily values of the CBL depth $\overline{z}_i$ averaged over 30-minute is shown in Fig. 4 for each of the 20 days. Fig. 4c, shows the average $\langle\overline{z}_i\rangle$ over the day. The maximum height of $\langle\overline{z}_i\rangle$ is $(1.53 \pm 0.1)$ km found at 13:30 UTC, which is about 2 hours after

local noon. The value at 13:00 UTC is very close to this maximum with $(1.52 \pm 0.1)$ km. In the course of the day there is a typical asymmetry: the growing phase from 06:00 to 13:30 UTC lasts longer than the declining phase from about 13:30 to 19:00 UTC. Interestingly, the standard deviations are typically smaller in the growing phase than in the decaying phase. We believe that this is related to more cumulus clouds at the top of the CBL in the afternoon.

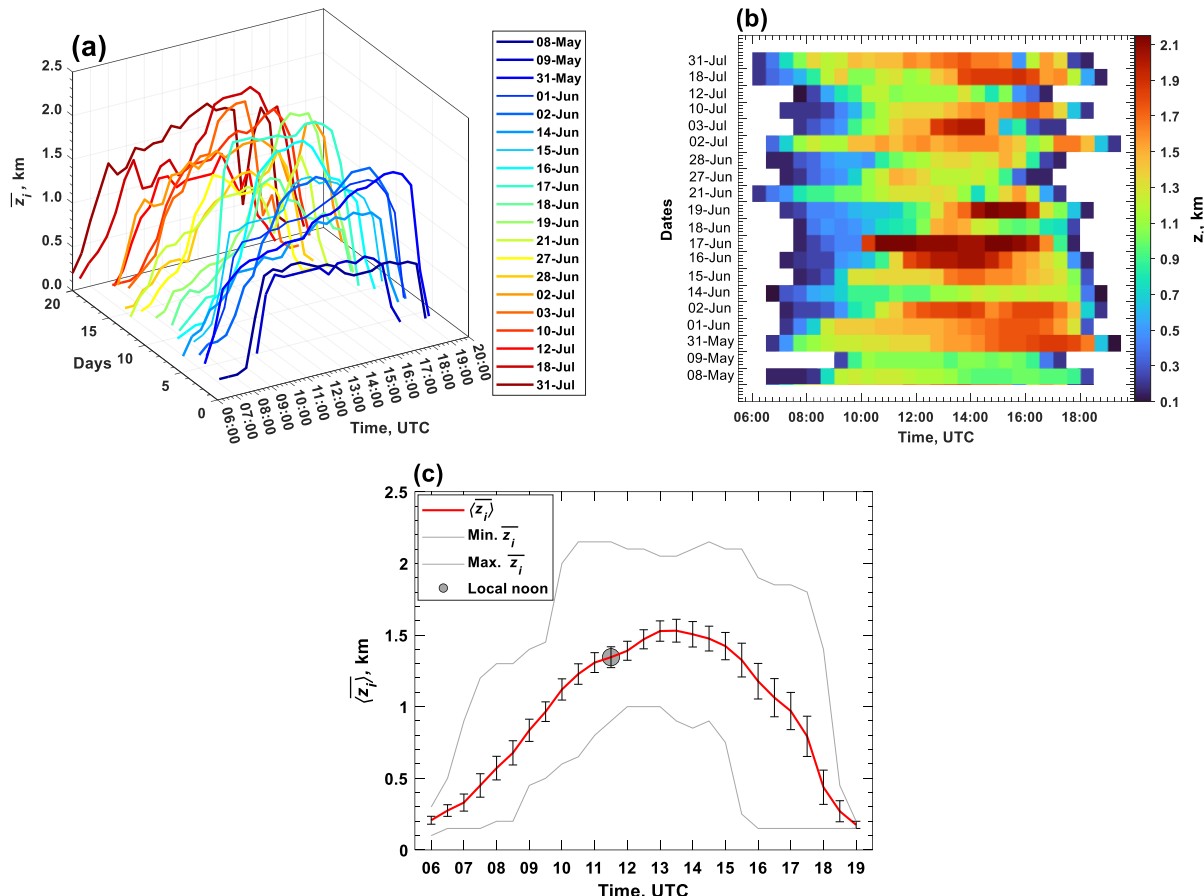


**Figure 4.** (a) 30-minute averages of the top height of the CBL $\overline{z}_i$. for all selected days. (b) same as (a) but as color plot (c) Mean of the data shown in (a) and (b) with standard deviations as well as minimum and maximum values. Local noon at 11:30 UTC is marked with a sphere.




## 4 Results and Discussion

### 4.1 Relationship between Surface Fluxes and Convective Boundary Layer Depth

We examined the relationships between the surface fluxes and CBL characteristics, which reflect the dynamics between land surface and atmosphere. Specifically, we show the co-evolution of $\langle \overline{z_i} \rangle$ and $\langle \overline{Q_n} \rangle$, $\langle \overline{S} \rangle$, and $\langle \overline{L} \rangle$ between 06:00 to 18:00 UTC (Fig 5).

In Fig. 5a, we can see that between 06:00 to 12:30 UTC, both $\langle \overline{Q_n} \rangle$ and $\langle \overline{z_i} \rangle$ rise steadily together in a linear way. During this interval, $\langle \overline{Q_n} \rangle$ increases from 32 to 661 Wm$^{-2}$ while $\langle \overline{z_i} \rangle$ increases from 0.20 to 1.47 km and we see a positive correlation coefficient during this time of 0.98. $\langle \overline{Q_n} \rangle$ reaches its peak at 12:30, but once that occurs, we observe a time lag of about 1 h before $\langle \overline{z_i} \rangle$ starts to decrease (at around 13:30 to 14:00 UTC). Then there is a decay phase between 14:00 to 18:00 UTC, when $\langle \overline{Q_n} \rangle$ decreases from 594 to 127 Wm$^{-2}$ and $\langle \overline{z_i} \rangle$ decreases from 1.50 to 0.43 km. The resulting relationship curve has a counterclockwise hysteretic pattern, i.e., where we can see a linear rise in $\langle \overline{z_i} \rangle$ with $\langle \overline{Q_n} \rangle$ but as $\langle \overline{Q_n} \rangle$ decreases again, $\langle \overline{z_i} \rangle$ does not reduce with it in a linear way. Similar hysteretic patterns between solar radiation and sensible heat were observed by Renner et al., (2019). The diurnal relationship between $\langle \overline{z_i} \rangle$ and sensible heat $\langle \overline{S} \rangle$ (Fig. 5b) exhibits somewhat similar behaviour to that of $\langle \overline{Q_n} \rangle$ but with an interesting difference. In the development phase, the relationship linear as before, with $\langle \overline{S} \rangle$ increasing from -0.71 to 88 W m$^{-2}$ while $\langle \overline{z_i} \rangle$ increases from 0.20 to 1.12 km between 06:00 to 10:00 UTC (correlation 0.97). Around local noon though (10:30 to 13:30 UTC) $\langle \overline{S} \rangle$ reaches a peak – somewhat earlier than $\langle \overline{Q_n} \rangle$. A maximum $\langle \overline{S} \rangle$ of 93 Wm$^{-2}$ is reached at 12:00 UTC. The resulting curve with $\langle \overline{z_i} \rangle$ is also a counterclockwise hysteretic pattern but with a more pronounced hysteresis pattern. This difference relates to the fact that the peak of $\langle \overline{S} \rangle$ occurs already at ~ 10.30 UTC (88 Wm$^{-2}$), only reaching 93 Wm$^{-2}$ at 12:00. In contrast, both $\langle \overline{Q_n} \rangle$ and $\langle \overline{L} \rangle$ (Fig. 5c) both continue to increase after that. $\langle \overline{L} \rangle$ increases to its maximum of 380 Wm$^{-2}$ at 11.30 UTC – ninety minutes after the peak of $\langle \overline{S} \rangle$.

In general, the very low Bowen ratio of 0.24 reflects overall the moist 'radiation-limited' regime in this period, and the high soil moisture availability. From 10:00, most of the energy is being partitioned into $\langle \overline{L} \rangle$ during the late morning, where the high $\langle \overline{Q_n} \rangle$ of 512 Wm$^{-2}$ allows the plants to strongly photosynthesize and hence freely transpire. This evapotranspiration produces a cooling effect, serving to inhibit $\langle \overline{S} \rangle$. At the same time, within complex feedback processes, entrainment of warmer drier air from the free atmosphere, due to vigorous convection can inhibit the sensible heat (by reducing the surface temperature gradient), whilst increasing $\langle \overline{L} \rangle$ by increasing the CBL moisture deficit. (Helbig et al., 2021).

The developmental phases of all relationships appear as one might expect. Incoming solar radiation heats both ground and subsequently the air above it (see $\langle \overline{T_{2m}} \rangle$ in Fig. 2f), with $\langle \overline{S} \rangle$ being dependent on the temperature gradient between surface and air above it. During the CBL development, the temperature gradient is larger, and $\langle \overline{S} \rangle$ and surface longwave emission transfers heat directly to the air and increases $\langle \overline{T_{2m}} \rangle$ linearly, and consequently TKE leads to increase the CBL during daytime in a linear manner. In the peak phase, the accumulated surface heating sustains strong convection which continued the growth of $\langle \overline{z_i} \rangle$



even after $\langle\overline{Q_n}\rangle$, $\langle\overline{S}\rangle$ and $\langle\overline{L}\rangle$ already reached the maximum values, and even start to decrease. These hysteresis effects between $\langle\overline{z_i}\rangle$ and $\langle\overline{Q_n}\rangle$, $\langle\overline{S}\rangle$ and $\langle\overline{L}\rangle$ are caused by the time dependent and nonlinear control of heat storage processes in the CBL (Renner et al., 2019). In the decay phase of CBL, $\langle\overline{Q_s}\rangle$ decreases and directly effect $\langle\overline{S}\rangle$ and leads to lowering the temperature gradient, consequently, reduction in turbulent mixing within CBL. The $\langle\overline{L}\rangle$ continues at a moderate level late into the afternoons remaining at ~225 Wm$^{-2}$ even at 18:00. This may be explained by the high evening temperatures of ~24 °C at 18:00 UTC (Fig. 2f). Together with the high soil moisture at Eddy Covariance 1 at this time (24% at 2cm, 37% at 15cm), we likely have good conditions for continued evapotranspiration from both soil and vegetation.

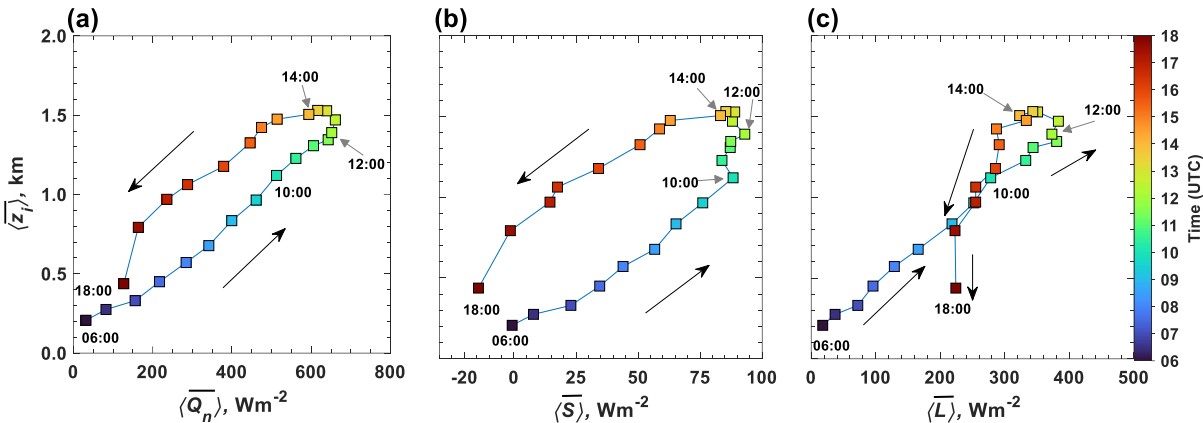

**Figure 5.** Hysteresis plots showing the relationships between (a) $\langle\overline{z_i}\rangle$ and $\langle\overline{Q_n}\rangle$, (b) $\langle\overline{z_i}\rangle$ and $\langle\overline{S}\rangle$, (c) $\langle\overline{z_i}\rangle$ and $\langle\overline{L}\rangle$ for 30-minute averages between 06:00 to 18:00 UTC. The upward black arrows show the development phase in the morning while downward arrows indicate the decaying phase of the CBL in the afternoon.

## 4.2 Analysis of Vertical Wind Variance and TKE Statistics

We have derived daily 30-minute profiles of $\overline{w'^2}$, $\overline{u'^2}$, $\overline{v'^2}$, and $\overline{TKE_{tot}}$ over all selected days between 06:00 to 18:00 UTC from May to July 2021. The vertical component of $\overline{TKE_{tot}}$ is denoted in the following as $\overline{TKE_V} = 0.5(\overline{w'^2})$, while the horizontal component of $\overline{TKE_{tot}}$ is $\overline{TKE_H} = 0.5(\overline{u'^2} + \overline{v'^2})$. We decided to use these parameters in order to compare the horizontal, vertical and total components of TKE quantitatively more easily. The spatial and temporal resolutions of $\overline{w'^2}$, $\overline{TKE_V}$, $\overline{TKE_H}$, and $\overline{TKE_{tot}}$ are 50 m and 30 minutes, respectively.

The profiles of $\overline{TKE_V}$, $\overline{TKE_H}$, and $\overline{TKE_{tot}}$ at 12:30 UTC as examples are shown in Fig. 6. The shape of the profile of $\langle\overline{TKE_V}\rangle$ is different to $\langle\overline{TKE_H}\rangle$ and thus to $\langle\overline{TKE_{tot}}\rangle$. The maximum values $\langle\overline{TKE_V}\rangle_{max}$, $\langle\overline{TKE_H}\rangle_{max}$, and $\langle\overline{TKE_{tot}}\rangle_{max}$ are 0.53 m$^2$s$^{-2}$, 1.17 m$^2$s$^{-2}$ and 1.60 m$^2$s$^2$ at $(0.31 \pm 0.06)\langle\overline{z_i}\rangle$, $(0.50 \pm 0.06)\langle\overline{z_i}\rangle$, and $(0.40 \pm 0.06)\langle\overline{z_i}\rangle$, respectively. Our





findings for $\langle\overline{TKE_V}\rangle_{max}$ agree with the results of previous studies (Lenschow et al., 2000; Dewani et al., 2023; Wulfmeyer et al., 2024).


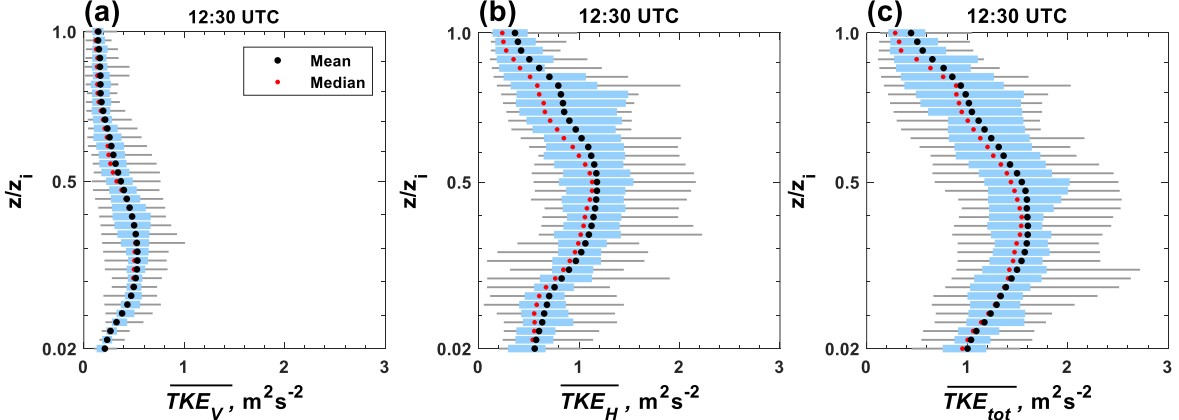

**Figure 6.** 30-minute mean profiles of (a) $\overline{TKE_V}$, (b) $\overline{TKE_H}$, and (c) $\overline{TKE_{tot}}$ at 12:30 UTC for the selected 20 days from May to July 2021. The mean and median values of this 20-day period at each height are marked with black and blue circles, respectively. Light blue boxes indicate the 25th-to-75th percentiles. Grey lines show the minimum and maximum values. The
local noon is at 11:30 UTC. The height of each profile is normalized to the CBL height $z_i$ of each data point before averaging.

Figures 7 and 8 show profiles and 3D time-height plots of $\langle\overline{w'^2}\rangle$ and $\langle\overline{TKE_{tot}}\rangle$, respectively. The sunlight heating the land surface triggers convection which leads to different values of $\langle\overline{w'^2}\rangle$, $\langle\overline{TKE_{tot}}\rangle$ and $\langle\overline{z_i}\rangle$ in the course of the day. In the morning hours between 09:00 to 11:00 UTC, $\langle\overline{w'^2}\rangle_{max}$ increases from 0.5 m²s⁻² to 0.94 m²s⁻² at heights of 0.36 km 0.45 km, respectively. At the same time, $\langle\overline{TKE_{tot}}\rangle_{max}$ increases from 0.89 m²s⁻² to 1.57 m²s⁻² at heights of 0.55 km and 0.70 km. The maximum values $\langle\overline{w'^2}\rangle_{max}$ and $\langle\overline{TKE_{tot}}\rangle_{max}$, are 1.1 m²s² and 1.7 m²s⁻² at 0.55 km and 0.70 km, respectively, at 12:00 UTC
while $\langle\overline{z_i}\rangle$ reaches its maximum of 1.53 km 1.5 hours later at 13:30 UTC. After reaching their maximum values, these quantities decline steadily towards sunset in response to reduced solar radiation. In the afternoon between 13:00 to 15:00 UTC, $\langle\overline{w'^2}\rangle_{max}$ decrease from 0.97 m²s² to 0.60 m²s² at constant height of 0.5 km, and $\langle\overline{TKE_{tot}}\rangle_{max}$ declines from 1.43 m²s² to 0.99 m²s² at 1.0 km to 0.9 km. Mostly, $\langle\overline{TKE_{tot}}\rangle_{max}$ is observed at different heights than $\langle\overline{w'^2}\rangle_{max}$.





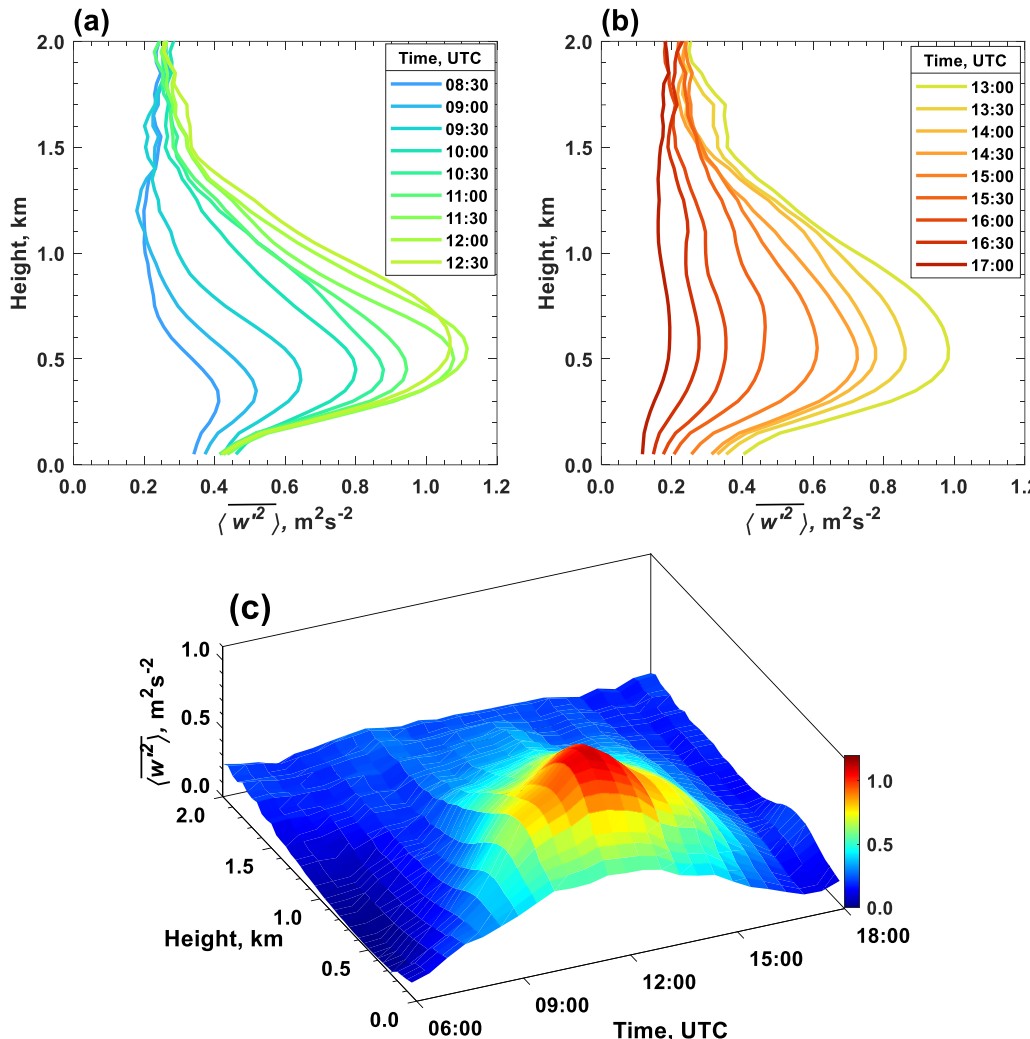


**Figure 7.** (a) Profiles of the vertical wind variance $\langle \overline{w'^2} \rangle$ in 30-minute windows averaged over the selected 20 days from 8:30 to 12:30 UTC. (b) same as (a) but for 13:00 to 18:00 UTC. (c) Same as in (a) and (b) but as 3D time-height plot.



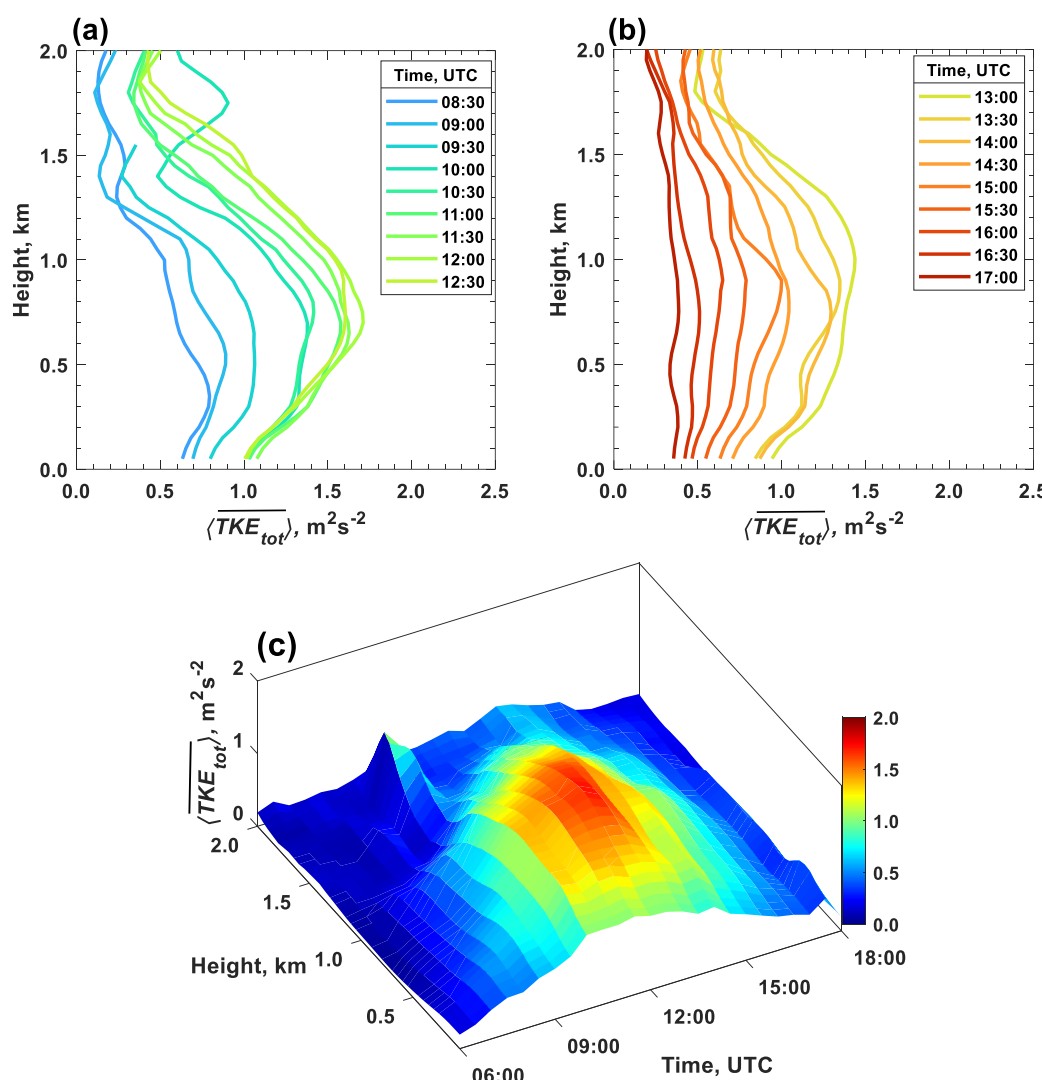

**Figure 8.** Same as Fig. 7 but for $\langle \overline{TKE_{tot}} \rangle$.



### 4.3 Relationships between Vertical Wind Variance, Horizontal Wind Variance and TKE

In Fig. 9, we present relationships between $\langle\overline{TKE_V}\rangle$ and $\langle\overline{TKE_{tot}}\rangle$, $\langle\overline{TKE_H}\rangle$ and $\langle\overline{TKE_{tot}}\rangle$, as well as $\langle\overline{TKE_V}\rangle$ and $\langle\overline{TKE_H}\rangle$
between 06:00 to 18:00 UTC. As expected, $\langle\overline{TKE_{tot}}\rangle$ is generally larger than $\langle\overline{TKE_V}\rangle$ for the same range and time as expected because the vertical wind variance is one component of TKE. Exemptions are due to noise. Generally, the relationships show linear trends in the morning and late afternoon hours. In these hours of the day, wind shear contributes more significantly than buoyancy generated turbulence in the CBL to TKE.

In the earlier morning (between about 06:00 to 09:00 UTC) and later afternoon (between about 15:00 to 18:00 UTC), we find
lower values. Here, $\langle\overline{TKE_{tot}}\rangle$ varies roughly between 0.1 m²s⁻² to 1.0 m²s⁻² while $\langle\overline{TKE_V}\rangle$ is about one-third as large.

Around local noon, the incoming solar radiation reaches its peak and generates - with a delay - large convective eddies which amplify vertical mixing. Consequently, around noon and in the early afternoon, namely between 11:00 and 14:00 UTC, we find the highest values of these variables with maximum values of 0.55, 1.2 and 1.7 m²s⁻² for $\langle\overline{TKE_V}\rangle$, $\langle\overline{TKE_H}\rangle$, and $\langle\overline{TKE_{tot}}\rangle$, respectively. In this peak phase, we find a different slope than before and after: changes of $\langle\overline{TKE_{tot}}\rangle$ are more strongly related
to changes of $\langle\overline{TKE_V}\rangle$ than to changes in $\langle\overline{TKE_H}\rangle$ in this phase of the day than before and after. Consequently, this shows that, in this peak phase of CBL turbulence, the vertical wind variance increases and decreases more strongly – on average – than the horizontal wind variance. Therefore between 11:00 to 14:00 UTC, vertical wind variance is larger than horizontal wind variance due to dominance of buoyant force over friction force. This relationship is also seen when comparing the horizontal wind variance $\langle\overline{TKE_H}\rangle$ and $\langle\overline{TKE_{tot}}\rangle$ (Fig. 9c and d) or $\langle\overline{TKE_V}\rangle$ and $\langle\overline{TKE_H}\rangle$ (Fig. 9e and f). In total, the resulting relationship
between $\langle\overline{TKE_V}\rangle$ and $\langle\overline{TKE_{tot}}\rangle$ tends to be non-linear. The same we found for the relationship between $\langle\overline{TKE_V}\rangle$ and $\langle\overline{TKE_H}\rangle$, while the relationship between $\langle\overline{TKE_H}\rangle$ and $\langle\overline{TKE_{tot}}\rangle$ seems approximately linear.






**Figure 9.** (a) Scatter plot of $\langle\overline{TKE_V}\rangle$ and $\langle\overline{TKE_{tot}}\rangle$ for 30-minute analysis periods for all selected days between 06:00 and 18:00 UTC. (b) Relative probability distribution of the data shown in (a). (c, d) and (e, f) Same as (a, b) but for $\langle\overline{TKE_H}\rangle$ and $\langle\overline{TKE_{tot}}\rangle$, as well as $\langle\overline{TKE_V}\rangle$ and $\langle\overline{TKE_H}\rangle$, respectively. A thin black line shows the identical function for comparison.



## 5 Summary and Outlook

We derived statistics of the relationship between turbulent variables in the convective boundary layer (CBL). For this, we used the data of two Doppler lidars, one in six-beam scanning mode, and other one in vertically pointing mode, during 20 days with cloud cover <40% at LAFO from May to July 2021. These data were combined with data of surface fluxes of an eddy covariance station. Using data of vertical pointing Doppler lidar, we derived $\overline{w'^2}$ profiles with 30 minute time windows and 50 m spatial resolution based on the methodology of Wulfmeyer et al. (2024). These profiles were determined from transverse

temporal ACF in the inertial subrange. From the six-beam scanning data, we obtained TKE and vertical wind variance. From these 30-minute data of each day, we derived statistics of 20 representative days characterizing well defined CBL.

By analysing the eddy covariance data, we have found that $\langle \overline{Q_n} \rangle$ followed a typical diurnal pattern being negative during night before sunrise and becoming positive in the daytime. It interestingly reached its maximum value at 12:30 UTC which is one hour after local noon. $\langle \overline{L} \rangle$ shows a similar kind of pattern. $\langle \overline{S} \rangle$ reaches a maximum already at 12:00 UTC. The Bowen ratio was

around 0.24 driven by a high soil moisture level in the analysis period with most of the energy being partitioned into $\langle \overline{L} \rangle$. $\langle \overline{T_{2m}} \rangle$ is found to be lowest just before sunrise, after which it increases and obtains a peak value at 16:00 UTC. The boundary layer height $\langle \overline{z_i} \rangle$ is determined by employing a fuzzy logic technique, using atmospheric vertical velocity variance profiles. The maximum $\langle \overline{z_i} \rangle$ of $(1.53 \pm 0.07)$ km is observed between 13:00 and 14:00 UTC, 2.5 hours after local noon and 1.5 to 2 hours after the maximum surface fluxes.

The relationships between $\langle \overline{z_i} \rangle$ and $\langle \overline{Q_n} \rangle$ as well as between $\langle \overline{S} \rangle$ and $\langle \overline{L} \rangle$ show counterclockwise hysteretic patterns. These hysteresis loops follow three phases of the CBL evolution: a development phase, a peak phase, and a decaying phase. In the development phase, $\langle \overline{z_i} \rangle$ increases approximately linearly with $\langle \overline{Q_n} \rangle$, $\langle \overline{S} \rangle$ and $\langle \overline{L} \rangle$. In the peak phase, $\langle \overline{z_i} \rangle$, $\langle \overline{Q_n} \rangle$, $\langle \overline{S} \rangle$ and $\langle \overline{L} \rangle$ show the largest values and do not change much. In the decaying phase, $\langle \overline{z_i} \rangle$ as well as $\langle \overline{Q_n} \rangle$, $\langle \overline{S} \rangle$ and $\langle \overline{L} \rangle$ decrease simultaneously in a non-linear way until sunset. The diurnal relationship of $\langle \overline{z_i} \rangle$ with $\langle \overline{S} \rangle$ shows a more pronounced hysteresis

loop due to a larger time lag as compared to the relationships of $\langle \overline{z_i} \rangle$ with $\langle \overline{Q_n} \rangle$ or $\langle \overline{L} \rangle$.

Furthermore, we analysed statistics of $\overline{TKE_V}$, $\overline{TKE_H}$ and $\overline{TKE_{tot}}$ in the CBL. We found maxima of $\langle \overline{TKE_V} \rangle$, $\langle \overline{TKE_H} \rangle$, and $\langle \overline{TKE_{tot}} \rangle$ at $0.30 \langle \overline{z_i} \rangle$, $0.56 \langle \overline{z_i} \rangle$, and $0.40 \langle \overline{z_i} \rangle$, respectively, just after local noon at 12:00 UTC. $\langle \overline{TKE_V} \rangle$ shows a non-linear relationship with $\langle \overline{TKE_{tot}} \rangle$ and $\langle \overline{TKE_H} \rangle$, while the relationship between $\langle \overline{TKE_{tot}} \rangle$ and $\langle \overline{TKE_H} \rangle$ is nearly linear.

In the future, we plan to extend our analyses of turbulent structures in the CBL under different meteorological conditions with

data of more years and at different sites. Thus, the instrumentation setup used in this study was proposed for the GEWEX Land–Atmosphere Feedback Observatories (GLAFOs; Wulfmeyer et al., 2020). We believe that such combinations of CBL statistics with surface flux measurements help to improve the representation of land-atmosphere interaction in numerical weather prediction models.




**Acknowledgments**

We gratefully acknowledge the Carl Zeiss Foundation for funding the setup phase of LAFO through its program to enhance research infrastructure at University of Hohenheim. We also like to thank Florian Späth for his support of the measurements.

**Code/Data availability**

Code/Data are available upon request from the corresponding author.

**Author contributions**

The turbulent measurement strategy has been designed and supervised by Volker Wulfmeyer and Andreas Behrendt. The codes

in IDL and Matlab for the data analysis were written by Syed Saqlain Abbas and Volker Wulfmeyer. The data analysis was mainly performed by Syed Saqlain Abbas, with contributions of Andreas Behrendt and Oliver Branch. The manuscript has been prepared and was written by all co-authors.

**Competing interests**

The contact author has declared that none of the authors has any competing interests.

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
