# Peer review of "Relationships Between Surface Fluxes and Boundary Layer Dynamics: Statistics at the Land-Atmosphere Feedback Observatory (LAFO)"

_EGUsphere, 2024_

## Author Comment (AC3)

**We thank both reviewers for their detailed comments, which have helped us to improve our manuscript. We have considered all comments and have revised the manuscript accordingly. In addition, we have made a few language corrections and refinements.**

**First Reviews:**

**a) Comments from Dr. David D. Turner:**

This paper analyzes 3 months of turbulence data (TKE, vertical velocity variance, and horizontal velocity variance) using observations from a flux tower at the LAFO site at the University of Hohenheim in southwestern Germany. Generally speaking, this paper is well-written and the results are both interesting and relevant. I am very supportive of the analysis of multi-month datasets to help understand what drives turbulent motions in boundary layer.

I have only a small number of comments, all of which should be addressed before the paper is accepted for publication.

First: while the abstract is clear, I think the title of the paper should be modified to state "Daytime" somewhere – perhaps "…and Daytime Boundary Layer…"

Thanks. We agree and change the title as suggested to "Relationships Between Surface Fluxes and Boundary Layer Dynamics During Daytime: Statistics at the Land-Atmosphere Feedback Observatory (LAFO)"

Second: given the physical arrangement of the LAFO site, it would seem that the measurements made by the DLs and flux station are really only connected if the wind direction was either from NNW-to-SSE or vice-versa; i.e., largely along the line between EC1 and the remote sensing site (Fig 1). However, no information was given on the mean wind direction in the CBL for the 20 days used in this analysis. If the analysis was restricted to winds that are largely along this axis, would the results be the same?

Thank you for this comment. We confirm that during these 20 days, the mean wind direction at 2 m height between 06:00 and 18:00 UTC was mainly between SE and SSE. We have added plots in the revised manuscript showing wind direction and speed (new Fig. 5).

Third: on line 197, the authors indicated they removed points that were 4 standard deviations away from the mean. This is a symmetric test; however, it is well-known that there can be significant skewness in vertical velocity profile (e.g., the Berg et al. 2017 paper that was referenced). Have the authors considered using an asymmetric test to identify outliers that preserves the skewness? This might have an impact on the variances they derive, which might be contributing to the occasional times when TKE_v > TKE_tot (line 375).

We fully agree. We have clarified in the revised version that we had determined the outliers considering this skewness (p. 10, l. 240–243).

Fourth: on line 207, the authors indicated that they used 15 lags for all of their autocorrelation analysis. This only works if the integral time scale is larger than 150 s for all of their cases; I would be surprised if this was true. They should only use lags from lag 1 to lag N where all N of the lags have significantly positive autocorrelations (not zero within uncertainty or negative). Would the authors please speak to this?

We thank the reviewer for pointing out this issue. This is typing error: we have used 1 to 5 lags not 15. We have corrected the text accordingly (p. 11, l. 255).

Fifth: it is not clear what scanning strategy is used for the 6-beam scanning approach in section 3.2. In particular, they indicate on line 226 that they are using the same approach to remove instrument noise from each of the beams (presumably they mean the AC method outlined in section 3.1); however, how was the integral scale (and hence number of lags) used here? I presume that at each beam angle, multiple 1-s samples were collected before the scanner moved to the next beam – however, I also assume that data was collect at each beam angle multiple times in the 30-min time window. More details are needed here to truly understand how they are doing their analysis.

We have employed a six-beam technique, as described by Sathe et al. (2015b), to estimate all the Reynolds stress tensor components. The Doppler lidar collects 20 six-beam scans in each 30-minutes time window. One full scan takes typically 90 s. Within each full scan, the DL records 10 measurements over a duration of 10 s at each position. To compute the variance of each beam, we have used the autocovariance at the first lag for the samples that were 1 s apart like Lenschow et al. (2000).

We have applied the higher-order moment analysis on the 1-s time series with considering changes in the scan direction. We are using lag 1 in the autocovariance analysis and there are 10 profiles from each direction. Only data of the same scan direction are combined in this analysis (same as in Bonin et al., 2017). We have added a comment on this in the text (p. 12, l. 271 & 279-280).

Lastly: in Figure 9, I believe they are showing the mean values throughout the CBL of TKE_v, TKE_tot, and TKE_h. Have the authors investigated the how these relationships change from the middle of the CBL (where all three of these terms are near their maximum values) vs near zi? In other words, would it be possible to break figure 9 into two figures; one to look at these scatter plots (relationships) between 0.3 to 0.6 zi, and other for 0.8 to 1.0 zi?

Thank you very much for this suggestion. In the revised version, we have separated the data in addition into two subranges: from 0.1 to 0.5 and from 0.5 to 1 zi, respectively (see new figures 12c, f, and i).

**Second Reviews:**

**b) Anonymous Referee**

The authors used data from two Doppler lidars and one eddy covariance station from the Land-Atmosphere Feedback Observatory (LAFO) in Germany, to investigate relationships between surface fluxes, CBL height, and profiles of vertical and horizontal wind variance and TKE. In total, 20 convective days from May to July 2021 were used. The days include cloud-free and cloud-topped CBLs.

Overall, the analysis and evaluations lag behind what has been provided so far in previous studies on that topic. For example, a solid comparison with results from previous studies (some possible studies for comparison are listed below) is missing. However, as the turbulence data in this study are not normalised by w*, a comparison is difficult anyway. Even the aim of the study is not clearly specified so that even the last sentence in the summary about future work is nearly consistent with the objective of this study. I suggest major modifications.

There seems to be a misunderstanding in the objective of our paper. We are discussing the vertical wind variance, TKE, and horizontal wind variances. These turbulent quantities are important for understanding the turbulent dynamics and could help to improve TKE parameterization in weather and climate models, pollutant dispersion, and wind energy studies. To the best of our knowledge, there is no previous study yet that presents real atmospheric measurements (and not only simulations) and discusses all of these quantities together as well as their relationships. As data, we are not just discussing a single case study but a large data set of 20 cases.

We discuss the relationships between mean surface fluxes and mean CBL height to analyse the correlation between them. It is true that we use data of one EC station – not (yet) several. However, we consider these data as representative for a larger region – as a large number of colleagues in doing in previous studies (e.g., Mauder et al., 2020; Dewani et al., 2023). The flux values in a region are closely correlated with each other – even though the

surface properties may vary; the Bowen ratio may vary because of different moisture, plants etc., the albedo may vary, ground heat flux may vary because of different ground properties. However, when analysing a larger number of cases, we believe that the relationships described here are representative for the unknown mean flux values with the exception of a scaling factor.

We have added some additional comment related to this point in the revised version of the manuscript (p. 3, l. 81-85).

Section 1 (Introduction)
General comments: The introduction is not well structured and should be modified.
Instead of citing the original papers about basic findings/knowledge on turbulence and ABL, quite new papers are referred to (so, at least cite the originals, too).

Even though we are not sure to which references the reviewer is referring to, we have added more references in the revised version of the manuscript, see also below.

The introduction contains several unnecessary repetitions.

Even though we are not sure to which text the reviewer is referring to, we have revised the text in order to avoid unnecessary repetitions.

Not all statements are precise or correct.

We are not sure to which text the reviewer is referring. Nevertheless, we have revised the text in order to make the text more precise and correct.

Just some examples of the above statements:
Line 22-23: "In daytime, the ABL becomes convective and turbulent processes dominate." By far not all ABLs are or become convective during daytime.
'During the daytime, the ABL is typically convective under clear-sky conditions; however, under strong synoptic forcing, such as high-pressure systems, convection may not occur.'"

We have changed the sentence to avoid any misunderstanding (p. 1, l. 24-26).

Line 24: …"with a morning transition to a maximum." To a maximum of what?

We have changed the sentence to clarify that we are referring to the height of the CBL top here.

Line 25 and 26: "Turbulence in the CBL is driven primarily by surface heating (Manninen et al., 2017), wind shear, and tropospheric entrainment (Wulfmeyer et al., 2016)."
This info is not new since 2016 or 2017 (as you suggest with these citations) but basic knowledge since half of a century (e.g. Kaimal et al 1976 or Caughey and Palmer 1979). So, you should primarily refer to the corresponding textbooks from e.g. Stull (1988: An introduction to the atmospheric boundary layer), Sorbjan (1989: Structure of the atmospheric boundary layer), Garratt (1994: The atmospheric boundary layer, Wyngaard (2010: Turbulence in the atmosphere) …… Lee (2023: Fundamentals of boundary-layer meteorology).

We fully agree and, certainly, we do not intend to say so. We have added the references as suggested by the reviewer to avoid any misunderstanding regarding this question. In addition, we keep the references to more recent studies because these studies utilize more advanced lidar technology and high-resolution data.

Concerning the conditions at the surface, soil and the energy balance equation of the Earth's surface (Munn, 1966; Oke, 1987; …… Arya, 2012 and Foken, 2017) you can also easily refer to the basics (i.e., radiation, soil heat flux, sensible and latent heat flux depend all on the conditions at the Earth's surface) – available in SVAT model descriptions.

We are not sure what basic publications the reviewer is referring to. We believe that a reference to the textbook of Foken (2017) is adequate and sufficient and it is clear that one cannot list all fundamental publications of a scientific area as references.

In the introduction the same information (repetition) is distributed over different lines like in line 28 "vegetation characteristics, and soil moisture", line 30 "and soil/vegetation fluxes".  Line 33: "transpiration of vegetation".

Thank you for these suggestions. We have revised the manuscript accordingly.

Repetitions: information in lines 59-63 (e.g. "The land surface properties influence the partioning of turbulent fluxes (Chen and Lo, 2023") is already mentioned several times before, e.g. in lines 20-36!

Thank you for these suggestions. We have revised the introduction to address the suggested needful changes for improving the overall structure and avoid repetitions.

In line 69 you mention that you use data from one EC station only instead of the two stations which are available. I don't know if you justify this in the following but with respect to surface and soil heterogeneity and corresponding spatial differences in fluxes, as mentioned in the previous part of the introduction, the restriction to one EC station only surprises me.

This is a misunderstanding. There is only high-quality data of one EC station – the one closer to the lidar site. There are unfortunately several gaps in the data of the other EC station further away from the lidar site so that we decided not to use these data. We have clarified this in the revised version to avoid this misunderstanding (p. 5, l. 131-142).

The aim of the paper (lines 71-72) is not very specific with respect to the problems raised above. E.g. precipitation in line 33. In lines 56-58 the energy balance closure problem is mentioned but is it part of the aim of this paper? Turbulence differences between cloud-free and cloud-topped CBLs. What exactly is intended to attack here? Parameterization? Model evaluation? Could you be more precise!

We understand the need for explaining the objectives of our work more clearly.
.

The objective is defined as investigating the statistics of CBL, $\overline{w'^2}$, TKE and surface fluxes within the CBL, and examining the relationship between surface fluxes and CBL variables.

This also highlight that these statistics are important for gaining a better understanding of land-atmosphere interactions and evaluating turbulence parameterization in mesoscale models.

**Section 2.1**

Line 82: If elevation is given in m AMSL please add AMSL

Thanks, we do so in the revised version. AMSL stands for above mean sea level.

Line 84: "As part of this synergy, Doppler lidars (Halo Photonics), a cloud Doppler radar (Metek GmbH), a water vapor differential absorption lidar (DIAL) (Späth et al., 2016), and a water vapor and temperature rotational Raman lidar (RRL) (Lange et al., 2019) are deployed to observe land surface fluxes …". How can you measure surface fluxes with the instruments listed in the first part of the sentence?

Doppler lidar and cloud Doppler radar do not measure surface fluxes directly but provide the essential wind data for flux measurement by performing near-surface scans.

Line 95: Specify "among other instruments" or delete.

Thanks, we specified the relevant instruments in the figure 1 caption.

Figure 1: could you add a figure which shows the surroundings of LAFO, e.g. where in Germany is LAFO situated.

Thank you for this suggestion. We have added a larger map in addition as suggested.

Line 84: why is the synergy of ….. the LAFO instruments unique? As far as I know there are several other similar observation platforms (Foken, 2021: atmospheric measurements, e.g. chapter 4.7). So, LAFO is one of these platforms and not unique.

The instrumentation and objective of LAFO are explained in detail in its overview paper (Späth et al., 2023), e.g., because of scanning water vapour and temperature lidars with worldwide unique accuracies and resolutions. We have clarified this in the text.

Line 106-107: repetition from the introduction. Delete!

Thanks, I have deleted these lines.

**Section 2.1.1**

General question: What are the measurement heights of the sensible and latent heat flux?

As usual, the measurements of sensible heat flux and latent heat flux are made at a height of 2 m above the canopy. We have added this information in the revised version.

Concerning radiation and soil heat flux: Figure 1 suggests that these measurements are made between the two fields. For which of the two vegetations are the radiation and soil heat flux measurements representative? E.g. albedo, surface temperature and soil moisture could be very different!

Indeed, the EC station is located between the fields as clearly shown in Fig. 1. Alfalfa and oats were cultivated in fields 4 and 5 during the period from May to July 2021, respectively. The radiation sensor was installed in Field 5 over this period.

We have removed the soil heat flux plot, as it is not discussed in this paper. However, in each of the field, three heat flux plates were installed at a depth of 8 cm to measure the soil heat flux. While computing the heat flux, the wind direction should be taken into consideration.

We assume that the radiation, albedo, and soil heat flux measurements are representative for the footprint of the flux measurements. Further details will be studied within upcoming measurements with fiber-optical distributed sensors.

It is the concept of LAFO, to place the EC stations in between the fields in order to investigate land-atmosphere feedback processes in complex terrain. Such complexity with neighbouring areas of different land use it typical for the region – as for many others in the world elsewhere. Please note that we analyze and discuss a larger set of days in order to obtain averages and representative statistics. We have added a footprint plot in the revised version and more information regarding this point in the revised version of the manuscript (p. 5, l. 135-144).

Line 120: How can the footprint always be within …. fields 4 and 5. For my understanding the footprint could be either within 4 or 5.

We have revised this sentence. The footprint between 06:00 to 18:00 UTC is nearly always within these fields during the analysis period. We have added a new figure (Figure 2) in the revised version to illustrate this. The flux footprint visualization was generated using the Kormann and Meixner (2001) footprint model in a Universal Transverse Mercator (UTM) coordinate system

**Section 2.1.3**

What is the vertical resolution of the radar data? Vertical extent, Temporal resolution?
The Doppler cloud radar (DCR) used in this study is a MIRA-36 by Metek GmbH, providing data with data resolutions of 1 s and 30 m up to 15 km.

This information has been added on line 171 of the revised version.

**Section 2.1.4**

Line 143-144: How did you calculate the cloud cover or better cloud fraction? Is it low, middle or high cloud fraction? Is it from temporal or spatial observations?

The cloud occurrence is estimated for all days though a procedure explained by Newsom et al., (2019b). The cloud base heights (CBH) are identified by the heights of sharp gradients in the 1 s range-corrected SNR profiles of the vertical pointing DL. It is applied to those profiles of SNR for which the values of the backscatter coefficient $\beta$ exceeds $10^{-4}$ $m^{-1}sr^{-1}$. After detecting the CBH, the corresponding profiles from lidar vertical wind data are removed from the CBH point up to 3 km top. CBH is both spatial and temporal observations. In the revised manuscript, we have added Figure 3, which includes (a) 10-minute cloud fraction and (b) 10-minute mean cloud cover between 06:00 and 18:00 UTC.
We have added this information in the revised version (p. 7, l. 170-180).

"days with cloud cover <40% during daytime, to ensure that the CBL was indeed developed by convection." Be careful with this statement. The role of ABL clouds on convection is not that clear. This is because CBL clouds might be a significant source convection due to elevated heat release, i.e. influencing the vertical wind variance and TKE in the upper part of the CBL. There are several studies on that topic, e.g. Garratt (1994), Neggers et al. (2003), e.g. Hogan et al. (2009), Chandra et al. (2010) and Lareau et al. (2018).

We fully agree with the reviewer and have revised this sentence to avoid any misunderstanding. In our study, we have used 20 days with maximum cloud cover 29.5 %. We have updated the manuscript to address these points more clearly in section 2.1.4.

Lines 144 and 154: Another example of repeating information: Line 144: "we selected only days with cloud cover <40% during daytime"
and
Line 154: "For our analysis, only those days are selected when cloud cover is <40%, otherwise the days were excluded." Even in this last sentence the information in the first part and second part of the sentence is redundant!
We agree and have edited the sentence accordingly in the revised manuscript.

Line 159ff: it would be good to define the different kinds of averages: bar and bracket. See also Eq. 8.

Thank you. We agree and have revised the manuscript to give a clear explanation of the different averages used in the study.

The description of the energy balance components is too superficial. Example: "In Fig. 2a and Fig. 2b, the mean net radiation $\langle Qn \rangle$ and the mean incoming solar radiation $\langle Qs \rangle$ follow typical diurnal patterns. Before sunrise, $\langle Qn \rangle$ is negative while $\langle Qs \rangle$ is zero. During daytime, both are positive."

Agreed. We aim to discuss the mean weather conditions, rather than focusing on individual days after sunrise between 06 to 18 UTC. We have changed the text and relevant figures (see Fig. 4 in the revised manuscript).

"In Fig. 2d, the mean of the latent heat flux $\langle L \rangle$ exhibits a similar pattern." Similar pattern in comparison to what? Compared to the sensible heat flux? The latent heat flux is not negative in the afternoon.

Thank you for your comment. Indeed, we are referring to the sensible heat flux here. We write "similar" and not "the same". But we agree that there could be a misunderstanding and have rewritten this point (p. 8, l. 204).

What I mean with superficial: the information which is of most interest is not given or discussed. This is - concerning the fluxes - what about the energy balance closure problem (mentioned in the introduction)? Does the imbalance depend on the wind direction? Are there differences from day to day?

We consider this fundamental problem of EC data as outside of the scope of this manuscript. We have added a comment in the revised version.

What are reasons/differences for times with sensible heat fluxes of about 20 W/m2 or 250 W/m2 during daytime? How do you explain latent heat flux values of nearly 200 W/m2 between 02 and 03 UTC or 21 to 24 UTC. Any explanation for the ground heat flux of about 130 W/m2 before sunrise?

There may be various reasons attributed to sensible heat fluxes of about 20 W/m2 or 250 W/m2 during daytime. It is 20 W/m2 due to thick clouds at the top of the CBL, wet soil due to rainfall, or calm wind reduce turbulence mixing, and horizontal transport of cooler air. Whereas its high value about 250 W/m2 can be associated with clear sky increase solar radiations, warm advection, and strong surface temperature.

However, the question regarding latent heat flux and ground heat are out of scope of our analysis periods. We have removed ground heat flux data in the revised manuscript to better align with our study objectives.

I would prefer to see and discuss diurnal cycles of wind speed and wind direction as well as cloud fraction of low, middle and high clouds in combination with the energy balance components and energy balance closure.

We agree that all these questions are very interesting; however, we consider these as outside of the scope of this work here.

E.g. it is essential for the CBL evolution to know whether the sensible heat is -50 W/m2 or +100 W/m2 at 15 UTC. What happens with the turbulence fluxes when the wind direction changes? Does the wind direction change during daytime at all? This would also have an effect on humidity and temperature.

In the revised version, we have added the statistics of wind speed (Fig. 5a), wind direction (Fig. 5b), and cloud fraction (Fig. 3a and 3b) as suggested by the reviewer. Please understand that individual days are not analyzed separately in our analysis. Our objective is to analyse mean statistics. We have extended the text and briefly describe these points in the revised version.

I also wonder about days when the 2 m temperature was 0 °C. Normally, the soil should be frozen on those days.

We agree that all these questions are very interesting; however, we consider these as outside of the scope of this work here. This data point was before 6 UTC and is no longer shown in the revised version of the manuscript. Nevertheless, we can confirm that indeed this value appeared once.

**Section 3.1**

Line 209: "The inertial subrange is the time interval in which the turbulence scales are locally homogeneous and isotropic within the CBL (Wulfmeyer et al., 2016)." Ones again: this is not a new finding from Wulfmeyer et (2016) but already stated by Obukhov (1941).

We fully agree and do not want to give the impression that this is a new idea. We have cited advance work presented by Wulfmeyer et al. (2016) because they utilized the high-resolution observational data to resolve inertial subrange and propose appropriate fit lags to study shape of autocovariance function. We have clarified this in the revised version.

Line 190 and 201: repetition: "known as autocovariance technique, is based on the fact that $w'(x, t)$ is correlated in time whereas $\varepsilon(x, t)$ is not." And "because noise is not correlated on time".

Thanks, we have revised the text accordingly.

General question and comment: When you use 30 min averages, you already cut the contribution of convective cells in the turbulence spectrum for periods of about ≥ 15 min. I expect that would be quite high when a larger convective cell dominates your 30 min time interval (see Figure 3, e.g. on 14 and 15 June). Have you look at that? Have you looked at the diurnal cycle of to see how much of the turbulence is cut by you short averaging time? This also holds for the calculation of TKE in section 3.2. If you consider Lenschow's "how long is long enough", your time interval could be too short (statistical problem). I don't know if these problems are discussed in the result sections.

We thank the reviewer for pointing this out. 30-minute averaging intervals are commonly used for turbulent studies in the CBL, while ensuring stationarity and isotropy in the high spatial and temporal resolution vertical wind velocity data (Sathe et al., 2015; Behrendt et al., 2020; Dewani et al., 2023; Wulfmeyer et al., 2024). We consider these intervals also suitable for our study. Statistical sampling uncertainties in single cases due to strong eddies are less relevant because we are discussing statistical averages of several cases. Furthermore, the sampling length in a given time interval depends on the horizontal wind. For the less extreme cases used in our study, we are considering a sampling interval of 30 minutes therefore as reasonable in order to keep the convective boundary layer turbulence as quasi constant in the analysis periods. We have added a comment in the revised version of the manuscript (p. 10, l. 235).

In Figure (a), we present the time series of 10 min average data of vertical wind velocity, which shows the longer-term variability and filtering out high-frequency fluctuations. In Figure (b), we show the 10 s average data, which keeps more of the high-frequency turbulent fluctuations.

This comparison illustrates how different averaging windows influence the representation of boundary layer turbulence.

[Figure]

**Section 3.2**

Eq. 8: Why are variances (bars) and covariances (bars and brackets) treated differently? i.e. different kind of averaging.

Thank you for pointing out the inconsistency regarding the averaging notations of variances and covariances. In Eq. 8 there was an error in the description of the averaging conventions used in the manuscript.

1. The overbar represents the 30-minute daily averages.
2. The angle brackets showing the mean values averaged over 30-minute intervals across all selected day.

We have corrected the typographical errors in the revised manuscript.

Line 226: Section 2.2.1 does not exist.

Thank you. Yes, this was a typographical error. The correct numbering is 3.1 instead of 2.2.1. We corrected this.

Line 210 and 234: The TKE-section and boundary layer depth-section are numbered equally: 3.2! The numbering of the sections is very sloppy. See comments before.

Thank you. We have corrected this.

Line 245: the criterion "fit best for our dataset." is a very subjective one. A more objective argumentation is needed here. Have you considered to use the noon-time radiosounding from Stuttgart for comparison.

Thank you. We agree and have revised the text. The radiosondes launched in Stuttgart by the German Weather Service are unfortunately too far away from our site and in different orographic terrain to allow comparisons in the CBL. The maximum of $\overline{w'^2}$ appeared to be at 0.30 to 0.4 $z_i$ – as has already been reported in previous scientific studies and provide an objective approach to validate $z_i$. We changed the revised manuscript accordingly (p. 12-13, l. 291-292).

Line 240: How can you state that your turbulence is surface based? "The maximum height of surface-based turbulence can be estimated". I hope you will discuss/explain the strong $z_i$-increase on 17 June (elevated turbulence) and on 19 June in connection with surface-based processes.

The discussion of single cases in outside of the scope of this manuscript. Of course, the boundary layer height depends on several factors, not just on local surface fluxes. We have revised the text accordingly in order to avoid misunderstanding (p. 13, l. 296-297).

To gain some insight into these day-to-day variations of zi, information on the wind speed and wind direction in the CBL could be helpful. Wind direction also could give hints concerning footprint of the convection measured by the wind lidars.

The discussion of single cases in outside of the scope of this manuscript. We have revised the text accordingly in order to avoid misunderstanding. We are showing these cases as examples.

**Section 4.1**

Line 295: How do you explain a CBL growth when the sensible heat flux is negative?

We are describing the measured data here. As written in the text: the sensible heat flux changes from negative to positive values in the morning. This allows the CBL to grow.

Line 304: "…. a cooling effect …" Where?

We rewrote the sentence to clarify this. This is kind of feedback, high evapotranspiration means more energy is used for latent heat flux and less energy for sensible heat flux, which consequently reduce surface heating. This feedback is stronger in well-watered, dense vegetated area.

305: "…. due to vigorous convection can inhibit the sensible heat ….". a) Do you mean "sensible heat flux"? 2b) what generates the vigorous convection when the sensible heat flux is quite low? Representative of the fluxes measured at EC1.

We accept that the original sentence in the manuscript may cause some misunderstanding to the readers. We have revised the sentence to better convey the clear meaning (p. 15, l. 343-345).

What we are saying here is that dry air from the free troposphere increases the demand of moisture in the CBL. This further increases the rate of evapotranspiration from surface and supresses sensible heat. CBL has a large heat capacity and stores significant amounts of heat for a longer period and gradually keep increasing the CBL height.

Lines 307-308ff: "Incoming solar radiation heats both ground and subsequently the air above it" It's the sensible heat flux divergence that heats the air above the ground and not the radiation!

We fully agree.

We have revised the sentence to clearly convey the meaning (p. 15, l. 346).

Line 308: Figure 2f does not show a temperature gradient – just the 2 m temperature.

Yes, indeed. We revised the text in order to avoid any misunderstanding (p. 15, l. 348).

Line 310: It would be helpful to add an equation which includes the physical processes you refer to when you write "consequently TKE leads to increase the CBL during daytime in a linear manner."

We acknowledge that describing TKE leads to increasing the CBL height in a "linear manner" may not be appropriate. We have revised the sentences (p. 15, l. 349).

Line 311: What do you mean with "accumulated surface heating"?

The heat buffer in the lower atmosphere near the land surface.

Line 314-315: To which levels do you refer to when you write "leads to lowering the temperature gradient"?

We have clarified this in the revised version. Turbulence is mainly driven by the temperature gradient near the surface due to incoming solar radiations.

General comment on Section 4.1:

(a) In order to see if your zi evolution is mainly based on the sensible heat flux and entrainment – as assumed for this investigation it would be helpful to check this. What would you expect concerning the zi-growth rate over homogeneous terrain when applying common models/equations describe the correlation between the sensible heat flux (Stull 1988) or Batchvarova and Gryning 1991) which? This would also help to understand some of the days with extreme (high and low) zi.

We thank the reviewer for this idea which we consider as highly interesting for future studies. However, we consider LES studies of homogeneous terrain as outside of the scope of this manuscript. We have added this point in the outlook.

(b) to investigate the mean behaviours of S and zi will always show a correlation. I guess that using the flux data from EC2 would result in a similar. However, to achieve a better knowledge of the dependence of zi on S, the understanding of the individual days would be helpful. Clustering concerning CBL clouds, CBL wind speed and direction.

The relationship between $\overline{S}$ and $\overline{z_i}$ is analyzed in the scatter plots for all selected days. This relationship shows significant day-to-day variability. We observe that the maximum relative probability density (Fig. b) appears around 09:00 to 12:00 UTC (Fig. a). This relationship is presented as the color bar, using daily 30-minutes average horizontal wind speed $\overline{V_s}$ (between 5 to 9 m s$^{-1}$) and wind direction $\overline{\theta_s}$ (between SE) in Figures c and d, respectively. To analyze the impact of boundary layer clouds on $\overline{S}$ and $\overline{z_i}$ relationship, we excluded two days (July 02, 31) when boundary layer clouds were present mostly near the top of the CBL. Similarly, in this case, the maximum relative probability density (Fig. f) likewise occurs between 09:00 to 12:00 UTC (fig. e), consistent with the pattern observed for all days.

In our case, when most of the days are clear but few days with cloud cover <30 %, $\overline{V_s}$ and $\overline{\theta_s}$ do not seem to have a notable impact on the observed relationship.

[Figure]

(c) What is the footprint for zi during daytime hours? How representative is the EC1? May the CBL be affected by the forest in the west?

For heterogeneous surfaces, it is very difficult to compute the horizontal extent of CBL footprint, depending on atmospheric conditions. The CBL footprint is coming from larger area, while EC station1 represent only a certain surface characteristics footprint near this flux station. However, the representative average flux values for the whole region are closely correlated and can be scaled with the fluxes of a single EC station in this region.

**Section 4.2**

Lines 334-335: "Our findings for $\langle TKE_V \rangle max$ agree with the results of previous studies (Lenschow et al., 2000; Dewani et al., 2023; Wulfmeyer et al., 2024). What do you mean with TKEmax agrees with previous results? As your values are not normalised by w* (e.g. Lenschow et al. (1980) and Sorbjan (1989), a comparison is not very meaningful.

In our study, we highlight the heights at which maximum values of $\langle TKE_V \rangle$ are observed. We have referenced three different studies, including Lenschow et al. (1980) and Wulfmeyer et al., (2024). These are the key references which provide a detailed analysis of $\langle TKE_V \rangle$ estimation by suggesting appropriate number of fit lags in the inertial subrange. However, normalizing $\langle TKE_V \rangle$ with w* can be a useful approach, it is beyond the scope of our study. We have clarified this in the text.

Lines 350: "The sunlight heating the land surface triggers convection …". Physically, it's the sensible heat flux that causes convection! E.g. if solar radiation is absorbed by the surface completely, convection would hardly be triggered at all. What I mean: be more precise with your statements!

Thank you for your highlighting the phrase. Yes, the more precise statement is: incoming solar radiation heats the surface, which in turn sensible heat flux increases near surface temperature and derive turbulence. We have revised the text accordingly to clarify this point.

General question:  You state in line 233 that you calculated w' based on the 6-beam scanning technique. In parallel, you received w' based on the vertically pointing lidar (section 3.1). What are the differences between both calculation of the vertical wind variance?

I would like to clarify that six-beam scanning DL provides 10 profiles in 10 s in one complete six-beam cycle with duration of about 90 s. But, from vertical pointing DL, we obtain continuous data of vertical wind with 1 s profiles. However, due to the higher resolution of the vertical pointing DL, we prefer to use the continuous vertical wind data obtained from the DL. We have clarified this in the text.

**Section 4.3**

Line 375-376: Are trivial sentences like "As expected, $\langle TKE_{tot} \rangle$ is generally larger than $\langle TKE_V \rangle$ for the same range and time as expected because the vertical wind variance is one component of TKE." really necessary?

Thank you. We have added "as expected" in the revised version of the manuscript.

Lines 380ff: here you start discussing the turbulence for different periods of the day based on Figure 9. However, in Figure 9 the data are not separated by hours but for 9-18 UTC only!

Figure 9 presents the mean diurnal relationship between $\langle TKE_{tot} \rangle$, $\langle TKE_V \rangle$ and $\langle TKE_V \rangle$ between 06:00 and 18:00 UTC with different colors for different times of the day We have clarified this in the revised version.

**Section 4.2, 4.3**

The analysis of the observations in general is quite superficial. Mean values (without clustering) do not really improve our knowledge of the CBL. This is one of the reasons why the authors do not provide more insight into CBL processes but just come up with such sentences in comparison to previous findings like "findings for $\langle TKE_V \rangle_{max}$ agree with the results of previous studies (Lenschow et al., 2000; Dewani et al., 2023; Wulfmeyer et al., 2024)." Only considering different conditions would have allowed to gain a deeper insight in the CBL processes.

In the present study, we clearly state that we are analysing a certain data set: cloud-free to weak-cloudy conditions at a certain site at a certain period. We do not claim that the results are representative for cloudy conditions, other regions, or other seasons. We fully agree that it would be very interesting to extend our study and added a statement in the discussion.

Understanding the turbulent dynamics under different weather conditions is indeed a good idea for future research studies.

A thorough comparison and discussion with results from e.g. Dewani et al 2013, Maurer et al 2016, Kiseleva et al 2024 could improve the benefit of this paper. For example, if days with cloud-topped CBLs and clear-sky CBLs would have been discussed separately (clustered), the observations could have provided a contribution to open question like the impact of CBL clouds on CBL turbulence (e.g. Hogan et al. (2009), Chandra et al. (2010) and Lareau et al. (2018). Another clustering would be CBL wind speed.

See previous reply. We agree that this will be very interesting and added the references suggested by the reviewer.

The scope of our study is quite different to the studies you mentioned. Dewani et al. (2023) has analysed the dependency of vertical velocity variance on meteorological conditions in the CBL using Doppler lidar measurements. Maurer et al. (2016) analysed dynamics of the CBL turbulent by focusing on vertical velocity variance profiles over six clear sky days. Lareau et al. (2018) study, has investigated the dynamics of shallow cumulus convection and highlights the importance boundary-layer factors which influencing the characteristics and behaviour of clouds.

**Miscellaneous and Typos**

If you use abbreviations for variables you should introduce the abbreviations when you mention them for the first time. When abbreviations are introduced for variables you should only use the abbreviations in the following (e.g. vertical wind variance in line 36 but the abbreviation is only introduced in line 45).

We agree and follow this idea. In line 36 we are introducing the vertical wind variance which includes instrumental noise as well. In line 45, the atmospheric vertical wind variance.

As mentioned, the numbering of the sections as well as referring to sections in the text is very sloppy. A little more care would make the reviewer's job easier! (See details above and below).

We agree and sorry for this. We have corrected the section numbering in the revised manuscript.

Line 50: "solar radiation are result of time" should be "solar radiation as result of time"

Thanks, we have corrected the text accordingly in the revised manuscript.

Line 61: partitioning in of portioning

Corrected.

Line 127: "2.1.3 Vertical Pointing and Six-Beam Scanning Doppler Lidars" This should be 2.1.2

Thank you. Corrected.

Line 147: "ms$^{-1}$." Without a blank, ms means millisecond! Use m s$^{-1}$ (occurs several times - check in the whole text!)

Thank you. Corrected.

Line 150: through instead of though?

Thanks, corrected.

Figure 2f: °C instead of C°

Thanks, corrected.

Line 196, 199, 204, 206 …… : "." Instead of "," because you start with a capital letter in the following line (197, 200, 205, 207).

Thanks, corrected.

Figure 4 (figure caption): ".…. of the CBL $\overline{z_i}$. for all selected days. (b) same as (a) but as color plot (c)": delete full stop after $z_i$ and add a full stop before (c). "same" should be "Same", i.e. use capital letter.

Thanks, corrected. Line 300: 11:30 instead of 11.30 UTC

Thanks, corrected.

Line 302: 10:00 should be 10:00 UTC

Thanks, corrected.

Eq. 9: normally, v' is defined as v' =  and not the other way around.

Thanks, corrected.

**Additional Language Refinements and Clarifications:**

We have included some additional language corrections and refinements in the revised manuscript to enhance clarity.

p. 1, l. 12; p. 3, l. 82-83; p. 16, l. 365-374.

**References**

Behrendt, A., Wulfmeyer, V., Senff, C., Muppa, S. K., Späth, F., Lange, D., Kalthoff, N., and Wieser, A.: Observation of sensible and latent heat flux profiles with lidar, *Atmos. Meas. Tech.*, 13, 3221–3233, https://doi.org/10.5194/amt-13-3221-2020, 2020.

Dewani, N., Sakradzija, M., Schlemmer, L., Leinweber, R., & Schmidli, J.: Dependency of vertical velocity variance on meteorological conditions in the convective boundary layer, J. Atmos. Chem. Phys., 23, 4045–4058, https://doi.org/10.5194/acp-23-4045-2023, 2023.

Kormann, R. and Meixner, F. X.: An analytical footprint model for non-neutral stratification, Bound.-Lay. Meteorol., 99, 207–224, https://doi.org/10.1023/a:1018991015119, 2001.

Mauder, M., Foken, T., and Cuxart, J.: Surface-energy-balance closure over land: a review, *Bound.-Layer Meteor.*, **177**, 395–426, https://doi.org/10.1007/s10546-020-00529-6, 2020.

Sathe, A., Mann, J., Vasiljevic, N., and Lea, G.: A six-beam method to measure turbulence statistics using ground-based wind lidars, *Atmos. Meas. Tech.*, 8, 729–740, https://doi.org/10.5194/amt-8-729-2015, 2015.

Wulfmeyer, V., Senff, C., Späth, F., Behrendt, A., Lange, D., Banta, R. M., Brewer, W. A., Wieser, A., & Turner, D. D.: Profiling the molecular destruction rates of temperature and humidity as well as the turbulent kinetic energy dissipation in the convective boundary layer, Atmos. Meas. Tech., 17(4), 1175–1196, https://doi.org/10.5194/amt-17-1175-2024, 2024.